# Instance-optimal Mean Estimation Under Differential Privacy

**Ziyue Huang, Yuting Liang, Ke Yi**
{zhuangbq,yliangbs,yike}@cse.ust.hk
Department of Computer Science and Engineering
Hong Kong University of Science and Technology

## Abstract

Mean estimation under differential privacy is a fundamental problem, but worst-case optimal mechanisms do not offer meaningful utility guarantees in practice when the global sensitivity is very large. Instead, various heuristics have been proposed to reduce the error on real-world data that do not resemble the worst-case instance. This paper takes a principled approach, yielding a mechanism that is instance-optimal in a strong sense. In addition to its theoretical optimality, the mechanism is also simple and practical, and adapts to a variety of data characteristics without the need of parameter tuning. It easily extends to the local and shuffle model as well.

## 1 Introduction

Mean estimation is one of the most fundamental problems in statistics, optimization, and machine learning. However, privacy concerns forbid us from using the exact mean in these applications, and the problem of how to achieve the smallest error under a given privacy model has received considerable attention in the literature. *Differential privacy* (DP) is a rigorous mathematical definition for protecting individual privacy and has emerged as the golden standard in privacy-preserving data analysis nowadays, which has been deployed by Apple [16], Google [22], and Microsoft [17].

Given a data set $\mathcal{D} := \{x_i\}_{i \in [n]} \subset \mathcal{U}^d$, where $\mathcal{U} = [u]$, i.e., each coordinate of the input vector is an integer (real-valued coordinates can be handled by quantization; see remark 1), our goal is to obtain a differentially private estimation $M(\mathcal{D})$ for the mean $f(\mathcal{D}) = \frac{1}{n} \sum_{i=1}^{n} x_i$ with small $\ell_2$ error $\|M(\mathcal{D}) - f(\mathcal{D})\|_2$. Because $f(\cdot)$ has global $\ell_2$ sensitivity $\mathrm{GS}_f = \sqrt{d}u/n$, the standard DP mechanism just adds Gaussian noise scaled to $\mathrm{GS}_f$ to each coordinate of $f(\mathcal{D})$, which results in an $\ell_2$ error proportional to $du/n$. This simple mechanism is worst-case optimal [28], but it is certainly undesirable in practice, as people often conservatively use a large $u$ (e.g., $u = 2^{32}$) but the actual dataset $\mathcal{D}$ may have much smaller coordinates. Instead, the *clipped-mean estimator* [1] (see Section 3.1 for details) has been widely used as an effective heuristic, but two questions remain unresolved: (1) how to choose the clipping threshold $C$; and (2) if it can yield any optimality guarantees. We answer these questions in a fairly strong sense in this paper.

### 1.1 Instance Optimality

As worst-case optimality is theoretically trivial and practically meaningless for the mean estimation problem when the global sensitivity is too large, one may aim at instance optimality. More precisely, let $\mathcal{M}$ be the class of DP mechanisms and let

$$\mathcal{R}_{\mathrm{ins}}(\mathcal{D}) := \inf_{M' \in \mathcal{M}} \inf\{\xi \mid \Pr[\|M'(\mathcal{D}) - f(\mathcal{D})\|_2 \le \xi] \ge 2/3\}$$

be the smallest error any $M'$ can achieve (with constant probability) on $\mathcal{D}$, then the standard definition of instance optimality requires us to design an $M$ such that

$$\Pr[\|M(\mathcal{D}) - f(\mathcal{D})\|_2 \le c \cdot \mathcal{R}_{\text{ins}}(\mathcal{D})] \ge 2/3 \tag{1}$$

for every $\mathcal{D}$, where $c$ is called the *optimality ratio*. Unfortunately, for any $\mathcal{D}$, one can design a trivial $M'(\cdot) \equiv f(\mathcal{D})$ that has $0$ error on $\mathcal{D}$ (but fails miserably on other instances), so $\mathcal{R}_{\text{ins}}(\cdot) \equiv 0$, which rules out instance-optimal DP mechanisms by a standard argument [21].

Since $\mathcal{R}_{\text{ins}}(\cdot)$ is unachievable, relaxed versions can be considered. The above trivial $M'$ exists because it is only required to work well on one instance $\mathcal{D}$. Imposing higher requirements on $M'$ would yield relaxed notions of instance optimality. One natural requirement is that $M'$ should work well not just on $\mathcal{D}$, but also on its neighbors, i.e., we raise the target error from $\mathcal{R}_{\text{ins}}(\mathcal{D})$ to

$$\mathcal{R}_{\text{nbr}}(\mathcal{D}) := \inf_{M' \in \mathcal{M}} \sup_{\mathcal{D}': d_{\text{ham}}(\mathcal{D}, \mathcal{D}') \le 1} \inf\{\xi \mid \Pr[\|M'(\mathcal{D}') - f(\mathcal{D}')\|_2 \le \xi] \ge 2/3\}.$$

Vahdan [35] observes that $\mathcal{R}_{\text{nbr}}(\mathcal{D})$ is exactly $\text{LS}_f(\mathcal{D})$, the *local sensitivity* of $f$ at $\mathcal{D}$, up to constant factors. However, $\text{LS}_f(\cdot)$ may not be an appropriate target to shoot at, depending on what $f$ is. For the MEDIAN problem, $\text{LS}_f(\mathcal{D}) = 0$ for certain $\mathcal{D}$'s and no DP mechanisms can achieve this error [33], while for mean estimation, $\text{LS}_f(\mathcal{D}) = \Theta(\text{GS}_f) = \Theta(\sqrt{d}u/n)$ for all $\mathcal{D}$, so this relaxation turns instance optimality into worst-case optimality.

The reason why the above relaxation is "too much" for the mean estimation problem is that $\mathcal{D}'$ may change one vector of $\mathcal{D}$ *arbitrarily*, e.g., from $(0, \dots, 0)$ to $(u, \dots, u)$. We restrict this. More precisely, letting $\text{supp}(\mathcal{D})$ denote the set of distinct vectors in $\mathcal{D}$, we consider the target error

$$\mathcal{R}_{\text{in-nbr}}(\mathcal{D}) := \inf_{M' \in \mathcal{M}} \sup_{\mathcal{D}': d_{\text{ham}}(\mathcal{D}, \mathcal{D}') \le 1, \text{supp}(\mathcal{D}') \subseteq \text{supp}(\mathcal{D})} \inf\{\xi \mid \Pr[\|M'(\mathcal{D}') - f(\mathcal{D}')\|_2 \le \xi] \ge 2/3\},$$

namely, we require $M'$ to work well only on $\mathcal{D}$ and its *in-neighbors*, in which a vector can only be changed to another one already existing in $\mathcal{D}$. Correspondingly, an instance-optimal $M$ (w.r.t. the in-neighborhood) is one such that (1) holds where $\mathcal{R}_{\text{ins}}$ is replaced by $\mathcal{R}_{\text{in-nbr}}$.

We make a few notes on this notion of instance optimality: (1) This optimality is only about the utility of the mechanism, not its privacy. We still require the mechanism to satisfy the DP requirement between *any* $\mathcal{D}, \mathcal{D}'$ such that $d_{\text{ham}}(\mathcal{D}, \mathcal{D}') = 1$, not necessarily one and its in-neighbors. (2) In general, a smaller neighborhood leads to a stronger notion of instance optimality. Thus, the optimality using in-neighbors is stronger than that using all neighbors, which is in turn stronger than worst-case optimality (i.e., $\mathcal{D}'$ can be any instance), while the latter two are actually the same for the mean estimation problem. (3) For an instance-optimal $M$ (by our notion), there still exist $\mathcal{D}, M'$ such that $M'$ does better on $\mathcal{D}$ than $M$, but it is not possible for $M'$ to achieve a smaller error than the error of $M$ on $\mathcal{D}$ over all in-neighbor of $\mathcal{D}$. This is more meaningful than ranging over all neighbors of $\mathcal{D}$, some of which (e.g., one with $(u, \dots, u)$ as a datum) are unlikely to be the actual instances encountered in practice.

## 1.2 Our Results

To design an $M(\mathcal{D})$ for the mean function $f(\mathcal{D}) = \frac{1}{n}\sum_{i=1}^{n} x_i$ that achieves an error w.r.t. $\mathcal{R}_{\text{in-nbr}}(\mathcal{D})$ for all $\mathcal{D}$, we need an upper bound and a lower bound. For the lower bound, we show that $\mathcal{R}_{\text{in-nbr}}(\mathcal{D}) = \Omega(w(\mathcal{D})/n)$, where $w(\mathcal{D}) := \max_{1 \le i < j \le n} \|x_i - x_j\|_2$ is the *diameter* of $\mathcal{D}$. Thus, from the upper bound side, it suffices to show that the mechanism's error is bounded by $c \cdot w(\mathcal{D})/n$. This is achieved in two steps. First, we use the clipped-mean estimator, but find the clipping threshold $C$ that optimizes its bias-variance trade-off, which is a certain quantile of the norms of the vectors in $\mathcal{D}$. However, we cannot use the optimal $C$ directly, as it would violate DP. Thus, we use a simple binary search based algorithm that can find any specific quantile privately with an optimal rank error. This results in a DP mechanism with error $\tilde{O}(\sqrt{d/\rho}) \cdot r(\mathcal{D})/n$, where $r(\mathcal{D}) := \max_i \|x_i\|_2$ and $\rho$ is the privacy parameter (formal definition given in Section 2). To reduce the error from $r(\mathcal{D})$ to $w(\mathcal{D})$, in the second step, we rotate and shift $\mathcal{D}$ into a $\tilde{\mathcal{D}}$ such that $r(\tilde{\mathcal{D}}) = O(w(\mathcal{D}))$ w.h.p., and apply the clipped-mean estimator (with our privatized optimal clipping threshold) on $\tilde{\mathcal{D}}$, leading to an error of $\tilde{O}(\sqrt{d/\rho}) \cdot w(\mathcal{D})/n$ for $n = \tilde{\Omega}(\sqrt{d/\rho})$. We also show that the optimality ratio $c = \tilde{O}(\sqrt{d/\rho})$ is optimal, i.e., any mechanism $M(\mathcal{D})$ having error $c \cdot w(\mathcal{D})/n$ for all $\mathcal{D}$ must have $c = \tilde{\Omega}(\sqrt{d/\rho})$ for $\rho < \tilde{O}(\sqrt{d/n})$.

Our mechanism has the following applications: (1) It can be applied directly to *statistical mean estimation*, where the vectors in $\mathcal{D}$ are i.i.d. samples from a certain distribution and one would like to estimate the mean of the distribution (in contrast, the version defined above is referred to as *empirical mean estimation*). For concreteness, we show how this is done for the multivariate Gaussian distributions $\mathcal{N}(\mu, \Sigma)$. For the case $\Sigma = \mathbf{I}$, our algorithm achieves an $\ell_2$ error of $\alpha$ using $n = \tilde{O}(\frac{d}{\alpha^2} + \frac{d}{\alpha\sqrt{\rho}})$ samples for $\alpha \leq O(1)$, matching the optimal bound in the statistical setting [8]. For a non-identity, unknown $\Sigma$, the error is proportional to $\|\Sigma^{1/2}\|_2$ as in [8]. Our mechanism requires only crude *a priori* bounds on $\mu$ and $\Sigma$ (i.e., the error depends on these bounds logarithmically), while [8] needs a constant-factor approximation of $\Sigma$, which can be obtained using $n = \tilde{\Omega}(d^{3/2}/\sqrt{\rho})$ samples [26]. Note that this can be a $\sqrt{d}$-factor higher than the sample complexity of mean estimation. Fundamentally, estimating $\Sigma$ is harder than estimating $\mu$, and we bypass the former so as to retain the same sample complexity of the latter. In practice, estimating $\Sigma$ first would consume the privacy budget from the mean estimation problem itself. On the other hand, the benefit of estimating $\Sigma$ first is that one can obtain an error guarantee under the Mahalanobis distance [26], which cannot be achieved by our method. (2) By simply changing the primitive operations, our mechanism easily extends to the local and shuffle model of differential privacy. In doing so, we also extend the one-dimensional summation/mean estimation protocol in the shuffle model [5] to high dimensions.

In addition to the theoretical optimality, our mechanism is also simple and practical. Most importantly, there is no (internal) parameter to tune. Yet, our experimental results demonstrate that our mechanism outperforms the state-of-the-art algorithm [8] with the best parameters tuned for each specific setting.

## 1.3 Related Work

Asi and Duchi [4] recently initialized the study on instance optimality under DP. They propose two ways to relax (equivalently, strengthen the requirement on $M'$) the strict instance optimality, which is unachievable. The first is to require $M'$ to be unbiased. This is not appropriate for mean estimation, since many estimators, including clipped-mean, is not unbiased. The second is to require $M'$ to work well over all the $r$-distance neighbors of $\mathcal{D}$ for $r \geq 1$. Thus, their optimality is weaker than using $\mathcal{R}_{\text{nbr}}(\cdot)$, hence not appropriate for the mean estimation problem (i.e., their optimality is the same as worst-case optimality). Instance optimality has not been studied in the local or shuffle model; existing protocols in these two models [7, 19, 5] all have errors proportional to the global sensitivity.

How to choose the clipping threshold $C$ for the clipped mean estimator has been extensively studied [2, 3, 34, 32], but existing methods do not offer any optimality guarantees. In particular, Andrew et al. [3] also use a quantile (actually, median) as $C$, but as we shall see, median is actually not the optimal choice. Furthermore, they use online gradient descent to find a privatized quantile, which does not have any theoretical error guarantees. Amin et al. [2] attempt to select an optimal quantile as the clipping threshold to truncate the number of contributions from each user, instead of clipping the actual samples in high dimensions as in our paper.

In the statistical setting, where the data are i.i.d. samples from some specific distribution, there are numerous methods [26, 8, 27, 29] that can avoid an error proportional to the global sensitivity, by exploiting the concentration property of the distribution. In particular, Biswas et al. [8] provide a simple and practical mechanism for multivariate Gaussian data. Levy et al. [30] propose a private mean estimator with error scaling with the concentration radius $\tau$ of the distribution rather than the entire range, but their algorithm requires $\tau$ to be publicly known in advance. In the local model, the algorithm in [23] uses a quantile estimation procedure based on binary search as a subroutine for one-dimensional Gaussian data.

Very recently, the relationship between the error of mean estimation and the diameter of the dataset has been exploited in [15] for low-communication protocols, but they do not consider privacy. Our DP protocols in the local and shuffle models have communication cost $\tilde{O}(d)$ per user (we do not state the communication costs in the theorems as they are not our major concern); it would be interesting to see if ideas from [15] can be used to reduce it further.

## 2 Preliminaries

### 2.1 Differential Privacy in the Central Model

**Definition 1** (Differential Privacy (DP) [21]). For $\varepsilon > 0$ and $\delta \geq 0$, a randomized algorithm $M : \mathcal{X}^n \to \mathcal{Y}$ is $(\varepsilon, \delta)$-differentially private if for any neighboring datasets $\mathcal{D} \sim \mathcal{D}'$ (i.e., $d_{\text{ham}}(\mathcal{D}, \mathcal{D}') = 1$) and any $E \subseteq \mathcal{Y}$,

$$\Pr[M(\mathcal{D}) \in E] \leq e^\varepsilon \cdot \Pr[M(\mathcal{D}') \in E] + \delta.$$

**Definition 2** (Concentrated Differential Privacy (zCDP) [10]). For $\rho > 0$, a randomized algorithm $M : \mathcal{X}^n \to \mathcal{Y}$ is $\rho$-zCDP if for any $\mathcal{D} \sim \mathcal{D}'$,

$$D_\alpha(M(\mathcal{D})\|M(\mathcal{D}')) \leq \rho\alpha$$

for all $\alpha > 1$, where $D_\alpha(M(\mathcal{D})\|M(\mathcal{D}'))$ is the $\alpha$-Rényi divergence between $M(\mathcal{D})$ and $M(\mathcal{D}')$.

Note that $(\varepsilon, 0)$-DP implies $\frac{\varepsilon^2}{2}$-zCDP, which implies $(\frac{\varepsilon^2}{2} + \varepsilon\sqrt{2\log\frac{1}{\delta}}, \delta)$-DP for any $\delta > 0$. To release a numeric function $f(\mathcal{D})$ taking values in $\mathbb{R}^d$, the most common technique for achieving zCDP is by masking the result with Gaussian noise calibrated to the $\ell_2$-sensitivity of $f$.

**Lemma 1** (Gaussian Mechanism [10]). *Let $f : \mathcal{X}^n \to \mathbb{R}^d$ be a function with global $\ell_2$-sensitivity $\text{GS}_f := \max_{\mathcal{D} \sim \mathcal{D}'} \|f(\mathcal{D}) - f(\mathcal{D}')\|_2$. For a given data set $\mathcal{D} \in \mathcal{X}^n$, the mechanism that releases $f(\mathcal{D}) + \mathcal{N}\left(0, \frac{\text{GS}_f^2}{2\rho} \cdot I_{d \times d}\right)$ satisfies $\rho$-zCDP.*

**Lemma 2** (Composition Theorem [10, 21]). *If $M$ is an adaptive composition of differentially private algorithms $M_1, M_2, \ldots, M_k$, then*

1. *If each $M_i$ satisfies $(\varepsilon_i, \delta_i)$-DP, then $M$ satisfies $(\sum_i \varepsilon_i, \sum_i \delta_i)$-DP.*

2. *For all $\varepsilon, \delta, \delta' \geq 0$, if each $M_i$ satisfies $(\varepsilon, \delta)$-DP, then $M$ satisfies $(\varepsilon', k\delta + \delta')$-DP, where*
$$\varepsilon' = \sqrt{2k\log\frac{1}{\delta'}}\varepsilon + k\varepsilon(e^\varepsilon - 1).$$

3. *If each $M_i$ satisfies $\rho_i$-zCDP, then $M$ satisfies $(\sum_i \rho_i)$-zCDP.*

### 2.2 Differential Privacy in the Local Model and Shuffle Model

The above definitions of DP and zCDP assume that $\mathcal{D}$ is handled by a trusted curator and only the output of the mechanism will be released to the public. Therefore, if the curator is corrupted, the privacy of all users will be breached. For weaker trust assumptions, the most popular models are the local model and the shuffle model, where each user holds their datum and locally privatizes (by some randomized mechanism) the message before sending it out for analysis. Hence, there is no third-party who has direct access to $\mathcal{D}$. Formally, each user holds one datum $x_i \in \mathcal{D}$, and the protocol interacts with the dataset using some local randomizer $R : \mathcal{X} \to \mathcal{Y}$, and the privacy guarantee is defined over the transcript (all messages sent during the protocol). For simplicity, we only present the definition for one-round protocols; the privacy guarantee of multi-round protocols can be composed across all rounds by the composition theorem. The definition below uses zCDP; other DP notions can be defined similarly.

**Definition 3** (Local Model (LDP)). A protocol using $R(\cdot)$ as the local randomizer satisfies $\rho$-zCDP in the local model if for any $x, x' \in \mathcal{X}$, any $\alpha > 1$, $D_\alpha(R(x)\|R(x')) \leq \rho\alpha$.

Due to the much stronger privacy requirement, the best accuracy guarantee of LDP protocols for several fundamental problems [12, 6, 18, 31] is a $\sqrt{n}$-factor worse than that in the central model. The shuffle model is established on an intermediary level of trust assumption between the local model and the central model and aims for obtaining errors closer to the central model. The key feature of the shuffle model is a trusted shuffler $\mathcal{S}$, which can permute all messages randomly before sending them to the analyzer, so that an adversary cannot identify the source of any message. Specifically, we consider the multi-message shuffle model, where each local randomizer $R : \mathcal{X} \to \mathcal{Y}^m$ outputs $m$ messages, and the transcript of the protocol $\Pi_P(\mathcal{D})$ is a random permutation of all $mn$ messages. The following definition uses $(\varepsilon, \delta)$-DP; the other two DP notions can also be defined similarly, but they do not offer the improvements that we want over LDP protocols.

**Definition 4** (Shuffle Model). A protocol $P$ satisfies $(\varepsilon, \delta)$-DP in the shuffle model if for any $\mathcal{D} \sim \mathcal{D}'$, and any set $E \subseteq \mathcal{Y}^{mn}$, $\Pr[\Pi_P(\mathcal{D}) \in E] \leq e^\varepsilon \cdot \Pr[\Pi_P(\mathcal{D}') \in E] + \delta$.

## 3 Our Method

### 3.1 Clipped-Mean Estimator

In the rest of the paper, we focus on the mean function $f(\mathcal{D}) = \frac{1}{n}\sum_{i=1}^{n} x_i$. Since $\mathrm{GS}_f$ is large, a very natural idea is to clip each vector in its $\ell_2$ norm by some threshold $C$. This reduces $\mathrm{GS}_f$ to $2C/n$, leading to the clipped-mean estimator [1]:

$$M_C(\mathcal{D}) = \frac{1}{n}\sum_{i=1}^{n} \min\left\{\frac{C}{\|x_i\|_2}, 1\right\} \cdot x_i + \mathcal{N}\left(\mathbf{0}, \frac{2C^2}{\rho n^2}\mathbf{I}\right). \tag{2}$$

**Lemma 3.** *For any given $C$, $M_C(\mathcal{D})$ satisfies $\rho$-zCDP, and has an expected $\ell_2$ error at most*

$$\mathsf{E}\left[\|M_C(\mathcal{D}) - f(\mathcal{D})\|_2\right] \le \mathcal{E}(C; \mathcal{D}) := \frac{1}{n}\sum_{i=1}^{n}\max\{\|x_i\|_2 - C, 0\} + \frac{C}{n}\cdot\sqrt{\frac{2d}{\rho}}.$$

An important remaining question is how to set the clipping threshold $C$. Setting it too low will result in a large bias, while setting it too high will introduce a large amount of noise. We show how to choose the optimal $C$ to balance this bias-variance trade-off. It is easy to see that the error $\mathcal{E}(C; \mathcal{D})$ is a convex function w.r.t. $C$, thus the optimal $C$ can be found by setting the derivative of $\mathcal{E}(C; \mathcal{D})$ to zero, i.e.,

$$\frac{\partial \mathcal{E}(C; \mathcal{D})}{\partial C} = \frac{1}{n}|\{i \in [n] \mid \|x_i\|_2 > C\}| - \frac{1}{n}\cdot\sqrt{2d/\rho} = 0.$$

Therefore, the optimal choice of $C$ is the $(n - \sqrt{2d/\rho})$-th quantile of $\{\|x_i\|_2\}_{i\in[n]}$.

### 3.2 Private Quantile Selection

However, we cannot use the optimal $C$ directly, as it would violate DP. Instead, we find a privatized quantile with small rank error. Specifically, for this problem, $\mathcal{D}$ consists of a sequence of ordered integers $0 \le x_{(1)} \le \cdots \le x_{(n)} \le u$. We would like to design a DP mechanism that, for a given $m$, returns an $x$ (which is not necessarily an element in $\mathcal{D}$) such that $x_{(m-\tau)} \le x \le x_{(m+\tau)}$[1] w.h.p. Here $\tau$ is referred to as the *rank error*. Existing methods on private range counting queries [11, 20] can be used for this purpose, but they actually find all quantiles, which is an overkill. Instead, we use a simple binary search algorithm [25, 14], which not only simplifies the algorithm, but also reduces the rank error (by polylog($u$) factors) to nearly optimal. Our algorithm `PrivQuant` makes use of a function `NoisyRC([a, b], D)` that returns a noisy count of $|\mathcal{D} \cap [a, b]|$, and we present the details in the supplementary material.

In the central DP model, we simply use `NoisyRC([0, mid], D)` $= |\mathcal{D} \cap [0, \text{mid}]| + \mathcal{N}(0, \log u/(2\rho))$.

**Theorem 1.** *The algorithm `PrivQuant` preserves $\rho$-CDP, and it returns a quantile with rank error $\tau$ with probability at least $1 - \beta$ for $\tau = \sqrt{\log u \log \frac{\log u}{\beta}/(2\rho)}$.*

In Section 4, we prove an $\Omega(\sqrt{\log u/\rho})$ lower bound (Corollary 1) on the rank error under zCDP for constant $\beta$. Thus the algorithm is optimal up to just an $O(\sqrt{\log\log u})$-factor.

We can now use `PrivQuant` to find an approximately optimal clipping threshold. Specifically, we invoke `PrivQuant` with $\rho' = \rho/4$ to find the $\max\{n - \max\{\sqrt{2d/\rho}, \tau\}, 1\}$-th quantile of $\{\|x_i\|_2^2\}_{i\in[n]}$. They are integers no more than $du^2$, so replacing $u$ by $du^2$ in Theorem 1 yields a rank error of $\tau = 2\sqrt{\log(du)\log\frac{\log(du)}{\beta}/\rho}$. Then we set $\tilde{C}$ as the square root of the returned quantile. Finally, we return the clipped mean estimator $M_{\tilde{C}}(\mathcal{D})$ with $\rho' = 3\rho/4$. The following theorem analyzes its error.

**Theorem 2.** *Our mean estimation mechanism is $\rho$-zCDP and has $\ell_2$ error $O(\sqrt{d/\rho} + \tau) \cdot r(\mathcal{D})/n$ with probability $1 - \beta$, where $\tau = 2\sqrt{\log(du)\log\frac{\log(du)}{\beta}/\rho}$.*

---

[1]Define $x_{(j)} = 0$ for $j < 1$ and $x_{(j)} = u$ for $j > n$.

### 3.3 Shifted-Clipped-Mean Estimator

To reduce the error from being proportional to $r(\mathcal{D})$ to being proportional to $w(\mathcal{D})$, we perform a random rotation on $\mathcal{D}$ followed by a translation. The rotation is done by $\hat{x}_i := HDx_i$, where $H$ is the Hadamard matrix, $D$ is a diagonal matrix whose diagonal entry is independently and uniformly drawn from $\{-1, +1\}$. Note that for now we omit the normalization coefficient $\frac{1}{\sqrt{d}}$ so that each coordinate of $\hat{x}_i$ is still an integer; we will apply the normalization to the final estimator instead. Then, for each $j \in [d]$, we invoke PrivQuant with $\rho' = \rho/(4d)$ to find an approximate median of $\{\hat{x}_i\}_{i \in [n]}$ along dimension $j$, denoted as $\tilde{c}_j$. Next, we shift the dataset to be centered around $\tilde{c} = (\tilde{c}_1, \ldots, \tilde{c}_d)$, obtaining $\tilde{\mathcal{D}} = \{\tilde{x}_i := \hat{x}_i - \tilde{c}\}_{i \in [n]}$. Note that $\tilde{c}$ has integer coordinates, so does $\tilde{x}_i$. Finally, we apply the clipped-mean estimator in Theorem 2 with $\rho' = \frac{3}{4}\rho$ on $\tilde{\mathcal{D}}$, obtaining an estimation $\tilde{y}$, and return $y := (\frac{1}{\sqrt{d}}HD)^{-1} \frac{1}{\sqrt{d}}(\tilde{y} + \tilde{c})$ as the mean estimator over $\mathcal{D}$.

**Theorem 3.** *Set* $\tau = \sqrt{\log(du) \log \frac{d \log(du)}{\beta}/\rho}$ *and assume* $n = \Omega(\tau\sqrt{d})$. *Our mean estimation mechanism is $\rho$-zCDP, and has $\ell_2$ error* $O\left(\left(\sqrt{d/\rho} + \tau\right)\sqrt{\log \frac{nd}{\beta}}\right) \cdot w(\mathcal{D})/n$ *with probability $1 - \beta$.*

**Remark 1.** For a dataset with real coordinates bounded by $R$ (in absolute value), one can quantize each coordinate to an integer using bucket size $\alpha/\sqrt{d}$, for any $0 < \alpha < R$, and then apply our algorithm over an integer universe of size $u = 2R\sqrt{d}/\alpha$. This just brings an additive $\alpha$ error to the error bound of Theorem 3.

**Statistical Mean Estimation.** Suppose $\mathcal{D}$ consists of i.i.d. samples drawn from the multivariate Gaussian distribution $\mathcal{N}(\mu, \Sigma)$, and we wish to estimate $\mu$, assuming *a priori* bounds $\|\mu\|_2 \leq R$ and $\sigma_{\min}^2 \mathbf{I} \preceq \Sigma \preceq \sigma_{\max}^2 \mathbf{I}$. We first clip each sample $x_i \leftarrow x_i \cdot \min\{R'/\|x_i\|_2, 1\}$ where $R' := R + 2\sigma_{\max}\sqrt{d + \log \frac{4n}{\beta}}$. Then we apply our mechanism with bucket size $\alpha/\sqrt{d}$ where $\alpha = \sigma_{\min}\sqrt{d/n}$. We analyze the error in the supplementary material. When $\Sigma = \mathbf{I}$ and ignoring $\log^{O(1)}(\frac{dnR}{\beta} \cdot \frac{\sigma_{\max}}{\sigma_{\min}})$ factors, the error in becomes $\tilde{O}\left(\frac{\sqrt{d}}{\sqrt{n}} + \frac{d}{\sqrt{\rho}n}\right)$, matching the known optimal bound for Gaussian mean estimation [8].

## 4   Lower Bounds

In this section we establish the instance optimality of Theorem 3 via three lower bounds: (1) $\mathcal{R}_{\text{in-nbr}}(\mathcal{D}) = \Omega(w(\mathcal{D})/n)$ for all $\mathcal{D}$; (2) an $\tilde{\Omega}(\sqrt{d/\rho})$ lower bound on the optimality ratio, and (3) that the condition $n = \tilde{\Omega}(\sqrt{d/\rho})$ is necessary.

The first lower bound follows from an observation by Vadhan [35]:

**Lemma 4** ([35]). *For any $f$, any $(\varepsilon, \delta)$-DP mechanism $M'$, and any neighboring datasets $\mathcal{D}_0 \sim \mathcal{D}_1$, there is a $b \in \{0, 1\}$ such that*

$$\Pr[\|M'(\mathcal{D}_b) - f(\mathcal{D}_b)\|_2 < \|f(\mathcal{D}_0) - f(\mathcal{D}_1)\|_2/2] < \frac{1 + \delta}{1 + e^{-\varepsilon}}.$$

**Theorem 4.** *For $\varepsilon < 0.1, \delta < 0.1$, $\mathcal{R}_{\text{in-nbr}}(\mathcal{D}) = \Omega(w(\mathcal{D})/n)$.*

The lower bound on the optimality ratio is by the reduction from statistical mean estimation [26].

**Theorem 5.** *Let $M$ be any $\rho$-zCDP mechanism for mean estimation that has $\ell_2$ error $c \cdot w(\mathcal{D})/n$ with constant probability for any $\mathcal{D} = \{x_i\}_{i \in [n]}$ drawn from $[u]^d$. If $\rho < \tilde{O}(\sqrt{d/n})$, then $c = \tilde{\Omega}(\sqrt{d/\rho})$.*

For the lower bound on $n$, we consider a weaker problem (so the lower bound is stronger), which is the $d$-dimensional version of the *interior point problem* [9]: Given a dataset $\mathcal{D} = \{x_i\}_{i \in [n]}$ drawn from $[u]^d$, the mechanism is only required to return a $y \in [u]^d$ such that $\min_i x_{ij} \leq y_j \leq \max_i x_{ij}$ for all $j$ with constant probability.

**Theorem 6.** *If there exists a $\rho$-zCDP mechanism that solves the interior point problem with success probability $2/3$, then $n = \Omega(\sqrt{d \log u}/\rho)$.*

Given a quantile selection mechanism with rank error $\tau$, by finding the median, the 1-dimensional interior point problem can be solved when $n = O(\tau)$. The following corollary then follows from Theorem 6.

**Corollary 1.** *Any $\rho$-zCDP mechanism for the quantile selection problem must have rank error* $\Omega(\sqrt{\log u/\rho})$.

## 5   Extension to the Local Model and Shuffle Model

Our mean estimation framework can be summarized as follows: (1) Given $\mathcal{D} = \{x_1, \ldots, x_n\}$, perform a random rotation, obtaining $\hat{\mathcal{D}} = \{\hat{x}_i := HDx_i\}_{i\in[n]}$; (2) For each $j \in [d]$, find an approximate median $\tilde{c}_j$ of $\hat{\mathcal{D}}$ along dimension $j$. Shift $\hat{\mathcal{D}}$ to be centered around $\tilde{c}$, obtaining $\tilde{\mathcal{D}} = \{\tilde{x}_i := \hat{x}_i - \tilde{c}\}$; (3) Find a clipping threshold $C$, which is the $m$-th quantile over the $\ell_2$ norms of the vectors in $\tilde{\mathcal{D}}$. In the central model, the optimal choice is $m = n - \sqrt{2d/\rho}$; (4) Perform $\ell_2$ clipping over $\tilde{\mathcal{D}}$ using $C$, and obtain a mean estimator $\tilde{y}$ of the clipped vectors. Finally, return $y = (\frac{1}{\sqrt{d}}HD)^{-1}\frac{1}{\sqrt{d}}(\tilde{y} + \tilde{c})$.

We note that each step has their counterparts in the local and the shuffle model: Step (1) is easy, where the randomized diagonal matrix $D$ can be generated using public randomness, or sent from the aggregator to each user if public randomness is not available. Step (2) and (3) both rely on quantile selection, which have alternatives in the local model and shuffle model. Step (4) is also easy since all vectors have norms bounded by $C$. We elaborate on the details in the supplementary material.

**The Local Model.** For step (4), the standard LDP mechanism is that each user applies the Gaussian mechanism (Lemma 1) with $GS = 2C$ to inject noise to their clipped vector, sends it out, and the aggregator adds them up and divides the sum by $n$. For quantile selection in step (2) and (3), we use the LDP range counting protocol in [13], which returns a data structure such that any range counting query can be answered. Putting things together, we obtain the following result.

**Theorem 7.** *Set $\tau = \sqrt{n/\rho}\log^2(du)\log\frac{du}{\beta}$ and assume $n = \Omega(\tau\sqrt{d})$. There is a 3-round $\rho$-zCDP mean estimation mechanism in the local model, achieving an $\ell_2$-error of $O\left((\sqrt{dn/\rho} + \tau)\sqrt{\log\frac{nd}{\beta}}\right) \cdot w(\mathcal{D})/n$ with probability $1 - \beta$.*

**The Shuffle Model.** We see that the error in the local model is worse than that in the central model by a $\sqrt{n}$-factor. It turns out that in the shuffle model, we can match the result in the central model up to logarithmic factors, albeit with $(\varepsilon, \delta)$-DP. This is mostly due to highly accurate summation and range counting protocols discovered recently for the shuffle model. We first extend the one-dimensional summation protocol in [5] for high-dimensional data by using random rotation.

**Lemma 5.** *Given $\mathcal{D} = \{x_i\}_{i\in[n]} \subset \mathbb{R}^d$ where $\|x_i\|_2 \leq C$ for all $i$, there is a one-round $(\varepsilon, \delta)$-DP mean estimation protocol in the shuffle model that returns a $y$ such that $\Pr\left[\|y - f(\mathcal{D})\|_2 \leq O\left(\frac{C}{\varepsilon n}\sqrt{\log(nd)\log\frac{d}{\delta}}\right)\right] \geq 2/3$.*

For the `NoisyRC` queries in the algorithm `PrivQuant`, we can use the range counting mechanism [24] in the shuffle model. Putting things together, we obtain the following result.

**Theorem 8.** *Set $\tau = \frac{1}{\varepsilon}\log^{3.5}(du)\sqrt{\log\frac{d\log(du)}{\delta}}$ and assume $n = \Omega\left(\tau\sqrt{d\log\frac{d}{\delta}}\right)$. There is a 3-round $(\varepsilon, \delta)$-DP mean estimation mechanism in the shuffle model, achieving an $\ell_2$-error of $O\left(\left(\frac{1}{\varepsilon}\sqrt{d\log\frac{d}{\delta}} + \tau\right)\sqrt{\log(nd)}\right) \cdot w(\mathcal{D})/n$ with probability $2/3$.*

## 6   Experiments

We performed both statistical and empirical mean estimation experiments to evaluate our method. For statistical mean estimation, we used multivariate Gaussian distributions with various $\mu$ and $\Sigma$. All algorithms are given the same $R, \sigma_{\min}, \sigma_{\max}$. We tried various $R$, while fixing $\sigma_{\min} = 0.1$ and $\sigma_{\max} = R/\sqrt{d}$. For empirical mean estimation, we used a real-world dataset, MNIST, which consists

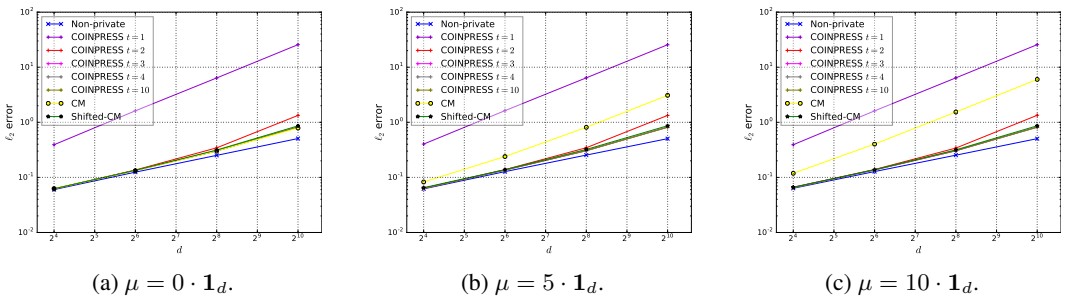

(a) $\mu = 0 \cdot \mathbf{1}_d$.        (b) $\mu = 5 \cdot \mathbf{1}_d$.        (c) $\mu = 10 \cdot \mathbf{1}_d$.

Figure 1: $\ell_2$ error vs. $d$ for $\mathcal{N}(\mu, I_{d \times d})$, where $n = 4000, \rho = 0.5, R = 50\sqrt{d}$.

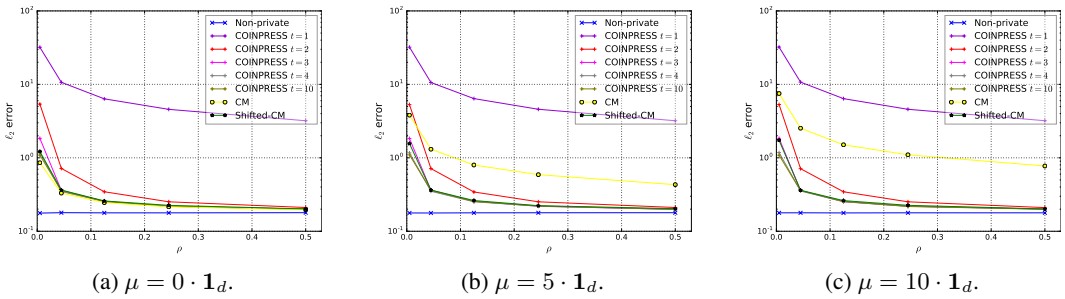

(a) $\mu = 0 \cdot \mathbf{1}_d$.        (b) $\mu = 5 \cdot \mathbf{1}_d$.        (c) $\mu = 10 \cdot \mathbf{1}_d$.

Figure 2: $\ell_2$ error vs. $\rho$ for $\mathcal{N}(\mu, I_{d \times d})$, where $n = 4000, d = 128, R = 50\sqrt{d}$.

of 70,000 images of handwritten digits, where each image is represented by a vector of dimension $d = 784 = 28 \times 28$. We quantized the values to integers $[u]$ for $u = 2^{10}$. We measured the $\ell_2$ error by taking the trimmed mean with trimming parameter 0.1 over 100 trials (as in [8]).

For the central model, we compared with COINPRESS[2] [8]. It starts with a given confidence ball of radius $R$ that captures the mean, and iteratively refines the confidence ball. The number of iterations $t$ is an important internal parameter in this algorithm; we tried $t = 1, 2, 3, 4, 10$ following their suggestion.

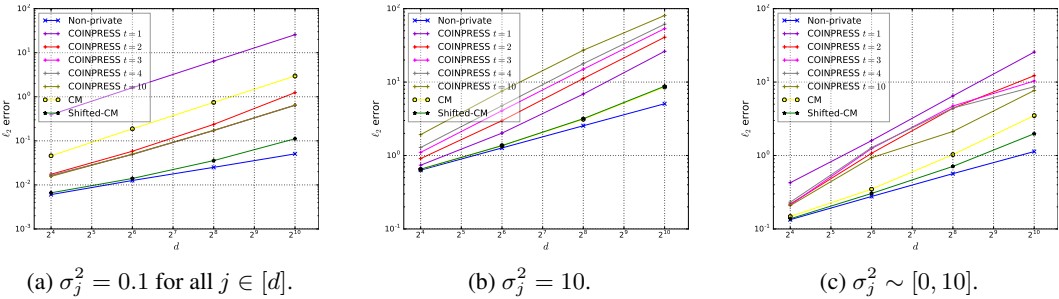

(a) $\sigma_j^2 = 0.1$ for all $j \in [d]$.     (b) $\sigma_j^2 = 10$.     (c) $\sigma_j^2 \sim [0, 10]$.

Figure 3: $\ell_2$ error vs. $d$ for $\mathcal{N}(\mu, \Sigma)$, where $\Sigma = \text{diag}([\sigma_1^2, \sigma_2^2, \ldots, \sigma_d^2]), n = 4000, \rho = 0.5, R = 50\sqrt{d}$.

The results for statistical mean estimation are shown in Fig. 1 to 5, where the detailed parameter settings are given in the captions. The results show that our method (Shifted-CM) performs at least as well as COINPRESS with the best $t$ across a variety of settings. In particular, when $\Sigma$ is not identity $I_{d \times d}$, our method significantly outperforms COINPRESS and offers errors close to the non-private estimator as demonstrated in Fig. 3 and Fig. 4. Moreover, as seen from Fig. 3, 4, and 5, the best choice of $t$ for COINPRESS is quite sensitive to $\Sigma$ and $R$, making it difficult to tune in practice.

---

[2]We also compared with another version of COINPRESS in the supplementary material, where we first scale the data by an estimated $\Sigma$ (obtained by MVCRec [8] using half of the privacy budget $\rho$).

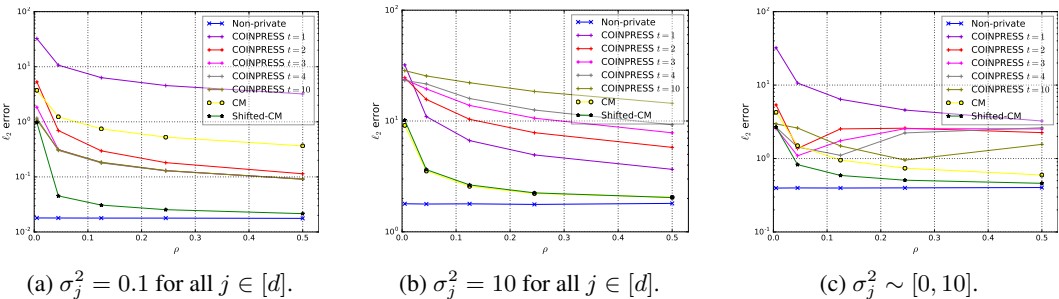

(a) $\sigma_j^2 = 0.1$ for all $j \in [d]$.  (b) $\sigma_j^2 = 10$ for all $j \in [d]$.  (c) $\sigma_j^2 \sim [0, 10]$.

Figure 4: $\ell_2$ error vs. $\rho$ for $\mathcal{N}(\mu, \Sigma)$, where $\Sigma = \mathrm{diag}([\sigma_1^2, \sigma_2^2, \ldots, \sigma_d^2]), n = 4000, d = 128, R = 50\sqrt{d}$.

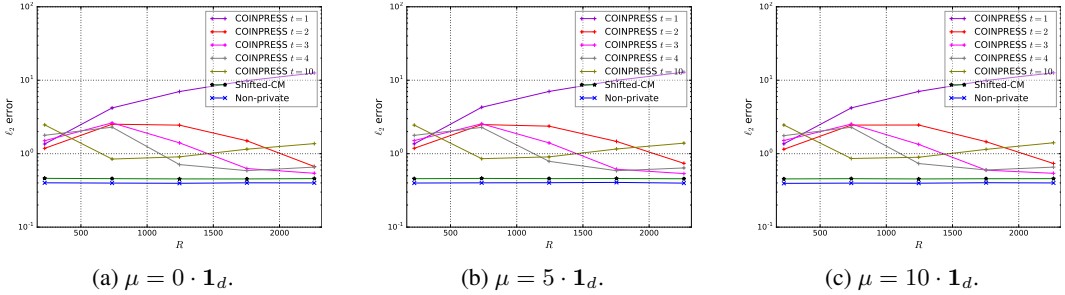

(a) $\mu = 0 \cdot \mathbf{1}_d$.  (b) $\mu = 5 \cdot \mathbf{1}_d$.  (c) $\mu = 10 \cdot \mathbf{1}_d$.

Figure 5: $\ell_2$ error vs. $R$ for $\mathcal{N}(\mu, \Sigma)$, where $\Sigma = \mathrm{diag}([\sigma_1^2, \sigma_2^2, \ldots, \sigma_d^2])$ and $\sigma_j^2 \sim_{\mathrm{u.a.r.}} [0, 10]$ for each $j \in [d]$, $n = 4000, \rho = 0.5, d = 128$.

Both our method and COINPRESS are translation-invariant. This can be verified from Fig. 1, 2, and 5, where the results are not effected by $\mu$. However, the approaches taken are different: COINPRESS uses an iterative process, while we shift the dataset to be centered around an approximate center point. In Fig. 1 and 2, we also include CM, our estimator without this shift operation, which is indeed affected by $\mu$.

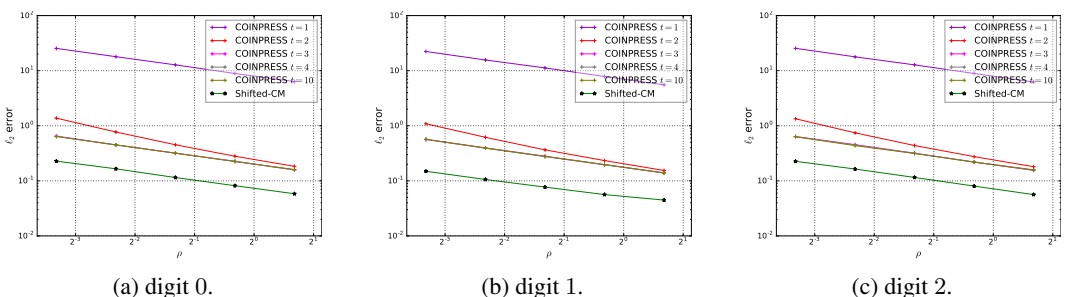

(a) digit 0.  (b) digit 1.  (c) digit 2.

Figure 6: $\ell_2$ error vs. $\rho$ for various digits in MNIST, where $d = 784, R = 50\sqrt{d}$.

For empirical mean estimation on the MNIST dataset, we see in Fig. 6 that our method outperforms COINPRESS for various privacy levels. This means that this dataset, as with most real-world datasets, does not follow Gaussian distribution with an identity $\Sigma$. The instance-optimality of our method is precisely the reason behind its robustness to different distributional assumptions, or the lack of.

One important component of our method is the optimal clipping threshold $C$, which is the $(n - \sqrt{2d/\rho})$-th quantile of the norms. Note that this depends on $d$ and $\rho$. Prior work [3] used a fixed quantile (e.g., the median). To better see the relationship between the optimal $C$ and $d, \rho$, we used a synthetic dataset $\mathcal{D} = \{i \cdot \mathbf{1}_d\}_{i \in [n]}$ with $n = 500$ and tried different quantiles as the clipping threshold. The results in Fig. 7 confirm our theoretical analysis: The optimal $C$ indeed depends on $d$ and $\rho$, while our choice attains the optimum.

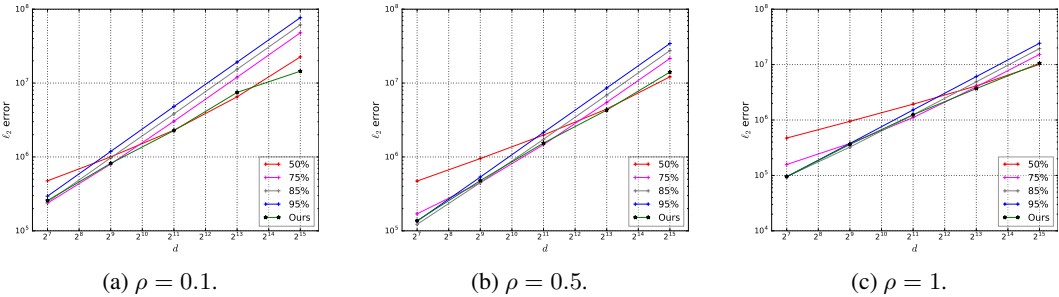

(a) $\rho = 0.1$.      (b) $\rho = 0.5$.      (c) $\rho = 1$.

Figure 7: $\ell_2$ error vs. $d$ for clipping at various quantiles of the norms on a synthetic dataset.

We also evaluated our algorithm in the local model and compare it with [23], and the results are presented in the supplementary material.

## Acknowledgments and Disclosure of Funding

We would like to thank Gautam Kamath and Jonathan Ullman for helpful discussions about Coin-Press [8].

Funding in direct support of this work: This work is supported by HKRGC under grants 16201318, 16201819, and 16205420.

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
