# OpenReview forum: "Instance-optimal Mean Estimation Under Differential Privacy"
_NeurIPS.cc/2021/Conference — NeurIPS 2021 Poster_

### Official Review · Reviewer_tbe2 · 2021-07-01

**Rating:** 6
**Confidence:** 4

**Summary:**

This paper does differentially private mean estimation for empirical estimation, as opposed to statistical estimation (statistical estimation is an application of their work). It focusses on instance optimal mean estimation by introducing a new target error function of the dataset, $\mathcal{R}_{\text{in-nbr}}$. It shows both and upper bound and a lower bound for this problem. It has both theoretical and empirical results.

The theoretical results say that for this measure of error, the optimal error scales according to the diameter of the dataset in question. To prove the upper bound, the authors first use a clipped-mean-estimator, with the clipping threshold decided via a simple derivative of the error. Since that clipping estimate is a function of the dataset, it could not be used directly. So, they came up with a simple quantile estimator to do the same. They, however, first performed a random rotation, followed by a translation to fix scaling issues. Then they finally use the aforementioned clipped-mean-estimator.

They finally showed application in statistical mean estimation of high-dimensional Gaussians. They showed that they require very weak prior knowledge of the covariance matrix of the Gaussian for this. They finally compared their empirical results with the Coinpress algorithm of Biswas et al.

They extend these results to the local and shuffle models of DP, too.

**Limitations And Societal Impact:**

Although the authors did not discuss any of this, I feel that there shouldn't be any negative impact of this work. This is a standard type of work in differential privacy literature, which can only be useful, if not harmful.

**Main Review:**

Update: Edited my final evaluation based on the authors' comments.

Strengths:
1. The paper has enough theory to support their claims, and enough empirical results on both real and synthetic datasets to verify the validity.
2. They achieve optimal error rates under their notion of error.
3. They achieve the same error for statistical mean estimation of high-dimensional Gaussians as Coinpress [Biswas et al.].
4. Their experimental results show that their algorithm works as well as (and in some instances, better than) Coinpress for a wide range of parameters. This is true for real datasets, too. So, their algorithms seem practical, just in terms of the error.
5. They also extend their results to the stricter models of differential privacy: local and shuffle models.

Weaknesses:
1. I don't understand why their error metric is the "right choice" for instance-optimal mean estimation. Like their algorithm implies good results for statistical mean estimation, but the metric looks a bit contrived. I didn't see any justification in their paper about this.
2. Regarding the choice of their error metric and the optimal error. The optimal error scales with the diameter of the dataset, which could well be $u\sqrt{d}$. Unless I'm missing something, this means that if most vectors in the dataset are "close to" $(0,\dots,0)$, and there is one anomaly of $(u,\dots,u)$, then the error would be pretty bad. It would be the same as adding noise scaled to the global sensitivity. So, how are they avoiding adding large amounts of noise here?
3. The choice of colours in the graphs is quite poor, which makes them hard to understand. Squinting too much while reading a paper is never pleasant.
4. It's nice that they were honest in the checklist, but the paper writing doesn't totally adhere to the prescribed format. Like they didn't discuss the limitations of their work. Is this because they were unaware of them? No code was provided either, so I'm just supposed to "trust their results" regardless of whether I want to run their code or not.
5. I wish they had reported the running times in their experiments, and compared with the state-of-the-art algorithms. It's hard to say how practical their work is without it.

**Time Spent Reviewing:**

4

---

> ### Author Response · Authors · 2021-08-09
> **Response to Reviewer tbe2**
>
> 1.
> We use the $\ell_2$ error, which we believe is the most natural metric for the mean estimation problem.
>
> 2.
> You are exactly right: If there is one anomaly vector $(u,...,u)$, the error of our algorithm will be $u\sqrt{d}$ (more precisely, it should be $u\sqrt{d}/n$).  However, any DP algorithm must incur error $u\sqrt{d}/n$ on this instance or its in-neighbor instance where all vectors are $(0,...0)$ (this follows from the instance-specific lower bound). Our algorithm chooses to have zero error on the all-0 instance (its diameter is 0), and (inevitably) incurs $u\sqrt{d}/n$ error on the instance with anomalies.  More generally, the entire line of works on "beyond global sensitivity", including ours, aim at achieving a graceful degradation as the hardness of the instances increases.  This is different from and more meaningful than worst-case optimal algorithm, which adds the same amount of noise $u\sqrt{d}/n$ to all instances (including the all-0 instance).
>
> Of course, instance-optimality is meaningful only under the assumption that typical instances do not have such anomalies.  Indeed, for the mean estimation problem, $u$ is usually very large (e.g., $2^{32}$) but the vectors we encounter in practice usually have small norms or close to each other (therefore having small diameter).  If you expect your instances to have anomalies (genuine inputs, not human/measurement errors), then simply using a worst-case optimal algorithm is already the best one can do.
>
> 4.
> We will release the code.
>
> 5.
> Same with Coinpress [8], we didn't report the running time, since both of our algorithms run very fast (a few seconds for each run) even on a laptop.  But we can certainly add a note in the paper mentioning this fact or giving some specific numbers.

---

> > ### Comment · Reviewer_tbe2 · 2021-08-09
> > **Response.**
> >
> > Comment 1: It was my misunderstanding of the metric. So, I take the blame for that.
> >
> > Comment 2: That makes sense. This work seems to help more in average-case data, which we're likely to expect.
> >
> > Overall: I'll bump my final evaluation of the paper.

---

> > > ### Author Response · Authors · 2021-08-10
> > > **Instance-optimality vs statistical setting**
> > >
> > > Thank you!
> > >
> > > With respect to comment 2: By "average-case" data, do you mean the statistical setting (i.e., data is drawn from some underlying distribution)?  Yes, both instance-optimality and the statistical setting look at "typical" data instead of the worst case, but the former is often stronger, because it gives a per-instance error guarantee.  At the end of Section 3, we show that an instance-optimal mean estimation algorithm with optimality ratio $O(\sqrt{d/\rho})$ implies an optimal mean estimation algorithm for Gaussian data (proof in supplementary material), but the opposite is not true, because it's OK for an algorithm in the statistical setting to do very badly on some instance, as long as that instance happens with low probability.

---

> > > > ### Comment · Reviewer_tbe2 · 2021-08-10
> > > > **Response.**
> > > >
> > > > Yeah, I meant "typical" instances. I get the difference between the two settings. Thanks for clarifying anyway!

---

### Official Review · Reviewer_aNGi · 2021-07-16

**Rating:** 6
**Confidence:** 4

**Summary:**

The paper studies instance-optimal mean estimation in the central model of differential privacy, with straightforward extensions to the shuffle and local models. The specific notion of instance optimality used is, I believe, original to this paper: the goal is to find an algorithm minimizing error over a given database D and a restricted neighborhood of databases that are neighbors and only include elements also present in D. The algorithm consists of four steps:

1) randomly rotate the database,
2) center it using a privately estimated component-wise median,
3) privately estimate a certain quantile of the L_2 norm of the resulting database and clip data to that estimated L_2 norm,
4) take a noisy empirical mean of the clipped data and de-bias.

The paper gives instance-specific upper and lower bounds that are nearly matching, up to logarithmic factors in the dimension and a discretization parameter. Experiments suggest that the algorithm performs well against existing methods.


**Limitations And Societal Impact:**

The paper flatly refuses to discuss potential negative societal impact. But that doesn't seem too unreasonable for something this abstract.

**Main Review:**

—Originality—

The most original theoretical component of the work might be its restricted-support definition of instance optimality. I don’t think I’ve seen that anywhere else. The algorithmic contributions are largely combinations of methods that have appeared elsewhere in the literature.

Two specific comments:
1) The discussion of other papers about the clipped mean estimator in the related work section is not quite accurate. For example, the Amin et al. paper not only uses a quantile as the clipping norm, but attempts to select an optimal one (not the median). Their setting is certainly different, since they are attempting to bound the number of contributions, not clip the actual samples, but the idea of truncating using an optimal quantile is there.
2) It would be interesting to see how a different private quantile algorithm affects the experimental results. As far as I know, the empirically strongest private quantile algorithms are based on the exponential mechanism (http://cs-people.bu.edu/ads22/pubs/2011/stoc194-smith.pdf, Algorithm 2), not binary search.

—Quality—

I did not check the proofs, but the algorithms and results seem believable, and proofs appear to be given when appropriate.

One omission: the submission does not provide any experimental code. Since I think the experiments are one of this paper’s strong point, the lack of code is disappointing.

—Clarity—

In general, the paper is well-written and easy to follow.

Some exceptions:
1) The initial presentation of the algorithm (erroneously, if I understand correctly) makes it sound like the L_2 clipping step is the first step of the algorithm. It wasn’t until I got to the section about local privacy that I realized I had it backward.
2) I think the lower bound section is missing a final statement that actually puts the separate lemmas together.
3) The experiment plots are hard to read. For example, the lines for Coinpress t= 4 and t = 10 are indistinguishable to me. Maybe make the lines thicker, or put some of these plots in the Appendix.
4) As far as I can tell, the main version of the paper doesn’t explain what the point of the random rotation step is. Some explanation appears in the proof of Theorem 3. I think this intuition should be more prominent.

—Significance—

To me, the strongest point of this paper is that the proposed algorithm, according to the experiments, achieves better results than the existing state of the art with fewer parameters to tune, and a pretty simple algorithm overall. I think the choice of things like clipping norms is a bit of a blind spot in the existing DP literature, so it’s good to see an application to a basic problem like mean estimation — it’s actually plausible that people would implement something like this in practice. The optimality result is also nice, though the proposed notion R_{in-nbr} does seem a little too restrictive to attach that much importance to it.

That said, the omission of experiment code and the criticisms above keep me from recommending the paper too highly. However, both of those things are fixable, and I look forward to the discussion period.

**Time Spent Reviewing:**

3.5

---

> ### Author Response · Authors · 2021-08-09
> **Response to Reviewer aNGi**
>
> --Originality--
>
> 1. Yes, Amin et al. also use the idea of selecting an optimal quantile as the clipping threshold, but the setting is different. We will add this reference in the final version if given the chance.
>
> 2. In fact, we considered the exponential mechanism and inverse sensitivity [1, 4] for the private quantile problem.  However, the binary search algorithm actually is better by a $\sqrt{\log u}$-factor (in terms of the rank error) in the CDP model as we can use the composition theorem of CDP.  The exponential mechanism satisfies pure DP, while its rank error is $\log u$ due to the utility theorem of the exponential mechanism (Corollary 3.12 in Section 3.4 in [A]). Moreover, the binary search algorithm can be easily extended to the local/shuffle model.
>
> [A] Cynthia Dwork, Aaron Roth, The Algorithmic Foundations of Differential Privacy
>
> —Quality—
>
> We will release the code.
>
> —Clarity—
>
> We will follow your suggestions in the final version if given the chance.

---

> > ### Comment · Reviewer_aNGi · 2021-08-15
> > **Re: Response to Reviewer aNGi**
> >
> > Re: point 2, I am suggesting that the exponential mechanism quantile algorithm from (http://cs-people.bu.edu/ads22/pubs/2011/stoc194-smith.pdf) may give stronger empirical performance than binary search methods, which would in turn perhaps boost the empirical performance of the method proposed in this submission. This is based on my own experience comparing private quantiles algorithms experimentally. That said, it's up to the authors whether they want to try that -- it won't affect my final review.

---

### Official Review · Reviewer_K1K1 · 2021-07-16

**Rating:** 6
**Confidence:** 3

**Summary:**

The paper proposes a new algorithm for the mean estimation problem that gets around the "curse of global sensitivity" and achieves the so-called instance optimality. Experiments including a real-data one are carried out to verify the practicality. The results are also extended to the shuffle and local models of DP.

**Limitations And Societal Impact:**

The authors have adequately addressed the limitations and potential negative societal impact of their work

**Main Review:**

It is always exciting to see that new DP algorithms can get around global sensitivity and most importantly, it works in practice and improves on previous algorithms.

I have one philosophical and one rigorousness/correctness criticism. Beginning with the philosophical one, the idea is that the error is a function of the dataset $D$, so why not compare it to something also depends on $D$. I agree with everything so far. My question is, why does comparing to the so-called "in-neighbors" make more sense than comparing to the diameter. The only appearance of the in-nbr after the introduction is to explain how it must be larger than the diameter. So it seems to me than in-nbr is just a story after the authors have the results regarding the diameter, which in my opinion is not more convincing than the diameter story.

Putting aside the philosophy, the in-nbr "shell" story may have some correctness issue. The main upper and lower bound seem to be Theorem 3 and 5 respectively. However, they are all stated in terms of the diameter. The statement about in-nbr is only partially supported by a one-sided inequality (Theorem 4). If in-nbr is much smaller than the diameter, then does your algorithm provide any guarantee w.r.t. in-nbr?

In my opinion, if the in-nbr thing is removed from the main story and treated as a potentially interesting connection, the paper is more correct and makes more sense.

Regarding the experiments, why not compare with other works in the line of "beyond global sensitivity"? Is coinpress the absolute best so far? More importantly, all the figures miss the comparison with the diameter $w(D)$, which is the main result. From the proof (if I understand correctly) for some specific non-Gaussian data (e.g. MNIST), the optimal factor can actually be smaller than the lower bound. If that's true, I wonder how much smaller the optimal factor can be.

Overall, I think it is a good and solid paper. However, I'd like to hear the explanation especially regarding the correctness issue before the final decision.

**Time Spent Reviewing:**

2.5

---

> ### Author Response · Authors · 2021-08-09
> **Response to Reviewer K1K1**
>
> The notion of in-neighbors is developed for our instance-specific lower bound $\mathcal{R}\_{\text{in-nbr}}(\mathcal{D})$, so it only appears in Section 1.1 and Section 4. As we explain in Section 1.1, the standard instance-optimality notion $\mathcal{R}\_{\text{ins}}(\mathcal{D})$ cannot be achieved by any DP algorithm, and the relaxation to the neighborhood-optimality considering all neighbors $\mathcal{R}\_{\text{nbr}}(\mathcal{D})$ [4] is too much (it is just worst case optimality for the mean estimation problem).
>
> > "The statement about in-nbr is only partially supported by a one-sided inequality (Theorem 4). If in-nbr is much smaller than the diameter, then does your algorithm provide any guarantee w.r.t. in-nbr?"
>
> This is perhaps a misunderstanding?  Theorem 4 is for the lower bound, so the one-sided inequality is enough, i.e., the instance-specific error bound ($\mathcal{R}\_{\text{in-nbr}}(\mathcal{D})$ by our definition) is at least the diameter of the dataset.
>
> We believe that Coinpress is the state of the art, which outperforms other prior works for DP mean estimation and is published at NeurIPS 2020.

---

> > ### Comment · Reviewer_K1K1 · 2021-08-23
> > **Still has correctness concern**
> >
> > Sorry I meant to say lower bound not algorithm. Let me ask the question again.
> >
> > Theorem 3 says if $n\geqslant \widetilde{\Omega}\left(\sqrt{\frac{d}{\rho}}\right)$ then
> > $$\mathrm{err}_{d,u,\rho,n}(D)\leqslant \widetilde{O}\left(\sqrt{\tfrac{d}{\rho}}\right)\cdot\frac{w(D)}{n}$$
> >
> > Theorem 5 says if $n\leqslant\widetilde{O}\left({\frac{d}{\rho^2}}\right)$ then
> > $$\mathrm{err}_{d,u,\rho,n}(D)\geqslant \widetilde{\Omega}\left(\sqrt{\tfrac{d}{\rho}}\right)\cdot\frac{w(D)}{n}$$
> >
> > I have two questions:
> > 1. Theorem 4 says $R_{in-nbr}(D)\geqslant {\Omega}\left(\frac{w(D)}{n}\right)$. I can only see how to use it to show
> > $$\mathrm{err}_{d,u,\rho,n}(D)\leqslant \widetilde{O}\left(\sqrt{\tfrac{d}{\rho}}\right)\cdot R\_{in-nbr}(D)$$
> >
> > What is the lower bound w.r.t. $R_{in-nbr}(D)$? By definition $\mathrm{err}_{d,u,\rho,n}(D)\geqslant R\_{in-nbr}(D)$? Do you claim that the upper and lower bound w.r.t. $R\_{in-nbr}(D)$ match? If so, why?
> >
> > 2. Theorem 5 requires $n$ to be small enough. Does that count as a valid lower bound? If $n$ cannot go to infinity, then there could be another algorithm that is significantly better than yours (which is never stated explicitly) in Section 3, even with respect to diameter.
> >
> > If the questions are not answered properly, then I cannot recommend an accept, as the introduction would be misleading and the main results are not clearly stated and more or less invalid.

---

> > > ### Author Response · Authors · 2021-08-24
> > > **About the lower bounds**
> > >
> > > Thank you for your questions on the lower bounds.  Your first question is w.r.t. the per-instance lower bound (Theorem 4) while the second one is on the lower bound of the (worst-case) optimality ratio $c$ (Theorem 5).  We answer each in turn:
> > >
> > > 1. The per-instance lower bound is $R\_{in-nbr}(D)$ and the upper bound is $c \cdot R\_{in-nbr}(D)$ for $c=\widetilde{O}\left(\sqrt{\tfrac{d}{\rho}}\right)$.  So we are not claiming that the upper and lower bound match, but with gap of $c$, called the optimality ratio.  Note that the optimality ratio in Asi and Duchi (ref [5] in paper) is $\tilde{O}(d/\varepsilon)$, which is even larger than ours.  In general, it's hard to match the per-instance lower bound, as we give too much power to the algorithm we compare against, which only needs to work well on the given instance (and its in-neighbors) and may fail miserably on other instances.  In some sense, instance optimality offers a way to measure an algorithm's performance against the hardness of the given instance ($w(D)$ in our case), which is much more meaningful than simply using the global sensitivity, which is indifferent to the hardness of the given instance.
> > >
> > > 2. You are right that any upper/lower bound must hold with asymptotic parameters for it to be meaningful, so $n$ must go to infinity.  Here, we consider all 3 parameters $n, d, 1/\rho$ to be asymptotic (i.e., they can all go to infinity); in fact, if $d,\rho$ are constants, then the target lower bound becomes $c\ge \Omega(1)$, which is trivially true.  Thus, the condition $n\leqslant\widetilde{O}\left({\frac{d}{\rho^2}}\right)$ does not mean that $n$ can only be a constant.  Instead, it poses a condition on the relative growth rates of the 3 parameters.  More precisely, if we unravel the asymptotic notation in Theorem 5, we obtain the following (ignore log factors): There exist universal constants $C_1, C_2$, such that for any $n, d, \rho$ that satisfy the condition $n < C_1 \cdot {d \over \rho^2}$, there exists a $d$-dimensional  instance $D$ of size $n$ on which any $\rho$-CDP mechanism must incur an error at least $C_2\cdot \sqrt{d\over \rho} \cdot {w(D) \over n}$ .
> > >
> > > Indeed, Theorem 5 does not rule out the possibility of achieving a better $c$ in the parameter regime $n\ge \widetilde{\Omega}\left({\frac{d}{\rho^2}}\right)$ (one can call it the large $n$, low dimension, or the low privacy regime, depending on the perspective).   Lower bounds that do not cover the entire parameter regime are actually not uncommon, since they are still useful in ruling out general results (i.e., it is not possible to achieve a better $c$ for all combinations of $n, d, \rho$).  Anyhow, we agree that the limitation of Theorem 5 should be stated more prominently in the introduction.
> > >
> > > Finally, we'd like to say that Theorem 5 is really a minor result of the paper, with a rather simple reduction proof (in the supplementary material).

---

> > > > ### Comment · Reviewer_K1K1 · 2021-09-01
> > > > **No correctness issue. Lower bound could be clearer.**
> > > >
> > > > I have tried by best to understand this $R\_{in-nbr}$ vs diameter result. Now there doesn't seem to be a correctness issue, but I still cannot convince myself why $R\_{in-nbr}$ makes so much sense that the introduction has to include it while excluding the much clearer diameter result. I would not dictate that the paper be written as I prefer, but the restriction on $n$ must be stated explicitly in the introduction. Because of the lack of clarity, I have to lower my score to 6.

---

> > > > > ### Author Response · Authors · 2021-09-01
> > > > > **Better clarity**
> > > > >
> > > > > We have to use $\mathcal{R}_{\mathrm{in\mbox{-}nbr}}(\cdot)$ in order to establish instance optimality. We actually follow the notation in [4], where they also use $\mathcal{R}(\cdot)$ to denote a neighborhood lower bound, but their neighborhood includes all neighbors.  We consider this as a major difference between [4] and our work, as including all neighbors to define instance optimality turns it into worst-case optimality for the mean estimation problem, so we included it in the introduction.  We agree that this makes the introduction less easier to understand than simply stating the diameter result.  We can perhaps follow your suggestion, stating only the diameter result in the introduction, while leaving its instance-optimality to a more technical section.
> > > > >
> > > > > We agree that it should be stated more explicitly that the lower bound on the optimality ratio $c = \sqrt{d /\rho}$ holds for $n\le O(d/\rho^2)$ (Theorem 5). Note that the upper bound holds for $n \ge \Omega(\sqrt{d/\rho})$.

---

### Official Review · Reviewer_Lthh · 2021-07-17

**Rating:** 6
**Confidence:** 4

**Summary:**

This paper considers the problem of private mean estimation and attempts to develop algorithms that are instance-optimal, that is, adapting to the difficulty of the underlying instance. The authors modify the existing instance-optimality definition in differential privacy, and develop an algorithm that is nearly instance-optimal. The caveat here is that nearly instance-optimal here hides factors that depend on the dimension and privacy parameters, hence making the optimality guarantee less strong than what claimed by the authors.

**Limitations And Societal Impact:**

yes

**Main Review:**

The paper has two main contributions: first, a new definition of instance optimality, and second a new algorithm for mean estimation that satisfies this definition of instance-optimality. I think the algorithm is nice and the experiments show some advantages in practice (although more experiments are needed; see below). However, I don't think the instance-optimality guarantees are strong enough as they hide a factor of c=\sqrt{d}/epsilon where d is the dimension. Hence it is misleading to say the algorithm enjoys strong instance-optimality guarantees when there is a sub-optimality factor of \sqrt{d}/epsilon, which are important problem parameters. More detailed comments below.

1. Instance-optimality: one of my main concerns about this paper is the fact that the algorithms are not really instance-optimal, having a sub-optimal factor of \sqrt{d}/epsilon which is an artifact of the way the authors define instance-optimality. Therefore I would first suggest the authors to clarify this in the paper. However, this doesn't rule out the possibility that the algorithm is really instance-optimal. It would help strengthen the paper if the authors show strong instance-optimality guarantees (up to constant factors) even in the 1-dimensional setting (using for example the instance-optimality notion of local minimax based on the hardest alternative of [1]). This will at least show that the algorithm has an instance-optimal dependence on the privacy parameter.

2. Clarity: overall the paper could use more work to make it more easily readable, especially in describing the algorithms; the authors currently write the algorithms in a standard text paragraph and I strongly recommend write them in the standard way as most papers do (in a box with pseudocode).

3. Binary search algorithm: the binary search algorithm may be losing a little because of it's iterative nature and a possible candidate here is the exponential mechanism\inverse sensitivity mechanism for quantiles [1,4].

4. Bound for Gaussian mean estimation: I like the bound for Gaussian mean estimation in corollary 1 in the appendix and I would recommend to have it in the main paper too as it shows some adaptivity to the covariance matrix. Some of the experiments were unnecessary and this bound can be added instead.

5. Experiments:  first, I suggest checking larger values of epsilon in the experiments to verify the same results hold in the low privacy regime. Moreover, the algorithms seems to have good performance in practice and so it is worth trying it with different applications of mean estimation such as DP-SGD. Finally, following on the comment above for instance-optimality at least in 1-dimensional setting, I recommend doing a 1-dimensional experiment in a similar setting to [3] and compare to the following baselines: the trimmed mean algorithm of [3] and the inverse sensitivity algorithm as in [2].



References

[1] Hilal Asi, John Duchi, Near Instance-Optimality in Differential Privacy, arXiv 2020

[2] Hilal Asi, John Duchi, Instance-optimality in differential privacy via approximate inverse sensitivity mechanisms, Neurips 2020

[3] Mark Bun, Thomas Steinke, Average-Case Averages: Private Algorithms for Smooth Sensitivity and Mean Estimation, Neurips 2019

[4] Jorg Drechsler et al., Non-parametric Differentially Private Confidence Intervals for the Median, arXiv, 2021


**Time Spent Reviewing:**

3-4

---

> ### Author Response · Authors · 2021-08-09
> **Response to Reviewer Lthh**
>
> 1. Indeed, there is a $\sqrt{d/\rho}$ gap between our upper bound and the instance-specific lower bound.  However, we show that this $\sqrt{d/\rho}$ optimality ratio is optimal. Please also see lines 32-34.  Optimality ratios for instance-optimal algorithms are akin to approximation ratios for approximation algorithms and competitive ratios for online algorithms.  All these instance-specific optimality notions try to identify some (instance-specific) lower bound OPT, e.g., the optimal solution found by a possibly exponential-time algorithm (for approximation algorithms), or the optimal solution found by the best offline algorithm (for online algorithms).  For our notion (as well as [1]) of instance-optimal DP algorithms, OPT is defined as the minimum error achievable by any DP mechanism designed specifically for the given instance and its neighbors.  Just like approximation algorithms and online algorithms, for many problems, OPT is not achievable due to the unreasonable power given to the algorithm we compare against.  But as long as we can show that the optimality/approximation/competitive ratio is the best one can hope for, the problem is considered to be well solved.  For instance, many approximation algorithms have approximation ratios on the order of sqrt(n) or higher (e.g., the best approximation ratio for MAX-CLIQUE is $O(n/\log^2 n)$, and it is known that the approximation ratio for any polynomial-time algorithm must be $\Omega(n^{1/2-\delta})$ for any constant $\delta$) .  In some sense, achieving OPT is not really the goal per se (as it is not achievable).  Instead, all these optimality notions try to measure the performance of the algorithm against the "hardness" of the instance, which are meaningful for problems where the instances have varying hardness and worst-case instances are atypical.  Finally, we would like to mention that the optimality in [1] depends on $O(d/\varepsilon)$ (even larger than ours), but importantly, they also prove that this factor is optimal under their notion of instance-optimality for pure DP.
>
> 2. We will add the pseudocode in the final version if given the chance.
>
> 3. In fact, we considered the exponential mechanism and inverse sensitivity [1, 4] for the private quantile problem.  However, the binary search algorithm actually is better by a $\sqrt{\log u}$-factor (in terms of the rank error) in the CDP model as we can use the composition theorem of CDP.  The exponential mechanism satisfies pure DP, while its rank error is $\log u$ due to the utility theorem of the exponential mechanism (Corollary 3.12 in Section 3.4 in [A]). Moreover, the binary search algorithm can be easily extended to the local/shuffle model.
>
> [A] Cynthia Dwork, Aaron Roth, The Algorithmic Foundations of Differential Privacy
>
> 5. For the tasks and the parameter settings in the experiments, we mostly follow Coinpress [8], but we can definitely also check larger values of eps.  In 1D, trimmed mean [3], inverse sensitivity [2], Coinpress, and our algorithm all have similar performance.

---

> > ### Comment · Reviewer_Lthh · 2021-08-16
> > **Response**
> >
> > I thank the reviewers for their detailed response. My main concern regarding instance-optimality remains unchanged. Please see comments below:
> >
> > 1. I don't find the comparison of the sub-optimality ratio in the paper to approximation ratios satisfying. In the case of approximation algorithms, the importance of these ratios is quite clear: this is the best approximation ratio that can be obtained using an efficient (polynomial time) algorithm. This also justifies having large ratios as otherwise an exponential runtime is required. On the other side, for the case of instance-optimality (and especially with a new definition proposed in the paper), justifying these sub-optimality ratios becomes less obvious, especially when these are too large. One of the goals of instance-optimality in general is to provide a different notion of performance that improves over worst-case optimality. Looking at the instance-optimality definition in the paper with large sub-optimality ratio ($c \approx \sqrt{d}/\varepsilon$), it is not clear why algorithms that are instance-optimal with this ratio are better than worst-case optimal algorithms (even though the sub-optimality ratio $c \approx \sqrt{d}/\varepsilon$ is optimal). For example, consider the simple setting of mean estimation where the optimal worst-case error is roughly $\frac{\sqrt{d}}{n \varepsilon}$. If we were to be satisfied with a minimal error of $\frac{1}{n}$ for any instance (or if this was indeed the minimal instance-specific error for any instance), then the sub-optimality ratio of any worst-case optimal algorithm is roughly $\sqrt{d}/\varepsilon$; the same as the algorithms in the paper. These comments and the fact that $\sqrt{d}/\varepsilon$ is the optimal ratio is somehow concerning as this implies that this notion of optimality is more similar to worst-case optimality than to instance-specific notions of optimality.
> >
> > 2. My comments in point 1 are the main reason that I suggested that the authors prove instance-optimality using the definitions in [1] (which only need small constant sub-optimality factors). Specifically, I propose to use the notion of local-minimax risk from [1] which builds on bounding the risk using the hardest alternative (this is different than the definition that the authors refer to in [2] which requires a large neighborhood).  Using this notion, the authors of [1] were able to show instance-optimality guarantees for 1-dimensional functions (with small constants). Proving similar guarantees is crucial to justify the instance-optimality of the algorithms in the paper (as claimed in the title).
> >
> > 3.  The problem with sub-optimality ratios that depend on the problem parameters (e.g. privacy) is that it makes it more challenging to determine which optimal algorithm is better in practice. For example, [1] show that for median estimation, the error of the instance-optimal algorithm (inverse sensitivity) is roughly $\frac{1}{n \varepsilon}$ and the error of the smooth sensitivity mechanism  is roughly $\frac{1}{n \varepsilon^2}$. This gap is also seen in practice where the former significantly outperforms the latter. Unfortunately, the definitions in this paper don'y allow to capture this; the optimal sub-optimality ratio is $1/\varepsilon$ (for $d=1$) hence both algorithms are instance-optimal according to this definition.

---

> > > ### Author Response · Authors · 2021-08-16
> > > **Response**
> > >
> > > > "For example, consider the simple setting of mean estimation where the optimal worst-case error is roughly $\frac{\sqrt{d}}{n \varepsilon}$. If we were to be satisfied with a minimal error of $\frac{1}{n}$ for any instance (or if this was indeed the minimal instance-specific error for any instance), then the sub-optimality ratio of any worst-case optimal algorithm is roughly $\sqrt{d}/\varepsilon$; the same as the algorithms in the paper."
> > >
> > > There might be some misunderstanding.  You seem to assume the data points are from the unit $\ell_2$ ball in your example.  In this setting, the worst-case optimal error $\frac{\sqrt{d}}{n \varepsilon}$.  However, the minimal error for an instance is not $\frac{1}{n}$, it is $\frac{w}{n}$, where $w$ is the diameter of the instance.  Thus, our error is $\frac{w\sqrt{d}}{n \sqrt{\rho}}$.  On the worst-case instance, $w=1$, but on most real-world instances, $w \ll 1$.  Thus, our error is always no worse than $\frac{\sqrt{d}}{n \varepsilon}$, but can be much better on typical instances.  In fact, the entire line of works on "beyond global sensitivity", including ours, aim at achieving a graceful degradation as the hardness (e.g., $w$ for the mean estimation problem) of the instances increases.
> > >
> > > > "My comments in point 1 are the main reason that I suggested that the authors prove instance-optimality using the definitions in [1]."
> > >
> > > For the mean estimation problem, the instance optimality of both [1] and [2] degenerates into worst-case optimality, as we explain in lines 41-45, 97-100.  To clarify more, the instance-specific lower bound in [1,2] is ${1 \over n \varepsilon}$, even in 1D.  Thus, the trivial, worst-case optimal algorithm is already instance-optimal under their notion of instance optimality.

---

### Decision · Program_Chairs · 2021-09-27

**Decision:**

Accept (Poster)

**Comment:**

This paper studies a fundamental problem in the differential privacy literature, namely mean estimation. It presents a novel algorithm and accompanies this with a theoretical and experimental evaluation. The reviewers all appreciate these contributions as valuable and worth accepting.

There are some reservations about the clarity of the paper. In particular, the algorithm is not clearly explained and the instance optimality claim hides a significant gap between the upper and lower bounds. Hopefully, these presentation issues can be resolved in the camera ready revision.